# Transcranial Direct Current Stimulation Enhances Cognitive Function in Patients with Mild Cognitive Impairment and Early/Mid Alzheimer’s Disease: A Systematic Review and Meta-Analysis

**DOI:** 10.3390/brainsci12050562

**Published:** 2022-04-27

**Authors:** Jiajie Chen, Zheng Wang, Qin Chen, Yu Fu, Kai Zheng

**Affiliations:** Department of Geriatrics, Tongji Hospital, Tongji Medical College, Huazhong University of Science and Technology, Wuhan 430074, China; d202081968@hust.edu.cn (J.C.); m201976230@hust.edu.cn (Z.W.); chenqin@tjh.tjmu.edu.cn (Q.C.); m202176106@hust.edu.cn (Y.F.)

**Keywords:** transcranial direct current stimulation, Alzheimer’s disease, mild cognitive impairment, cognitive function, meta-analysis

## Abstract

Transcranial direct current stimulation (tDCS) i a non-invasive brain stimulation which is considered to have the potential to improve cognitive impairment in patients with mild cognitive impairment (MCI) and Alzheimer’s disease (AD). However, previous studies have been controversial on the therapeutic effect of tDCS. This meta-analysis aimed to evaluate the effects of tDCS on cognitive impairment in patients with MCI and mild-to-moderate AD. Five databases, namely PubMed, EMBASE, MEDLINE, Web of Science and The Cochrane Library, were searched with relative terms to extract the cognitive function changes measured by an objective cognitive scale in the included studies. The meta-analysis results showed that, compared with sham tDCS treatment, the overall cognitive function of patients with AD and MCI was significantly improved (weighted mean difference = 0.99; 95% confidence interval, 0.32 to 1.66; *p* = 0.004) after tDCS treatment, but the behavioral symptoms, recognition memory function, attention and executive function were not significantly improved. The subgroup analysis showed that the treatment would be more efficacious if the temporal-lobe-related brain areas were stimulated, the number of stimulations was greater than or equal to 10 and the current density was 2.5 mA/cm^2^. Among them, AD patients benefited more than MCI patients. No cognitive improvement was observed in patients with MCI or AD at different follow-up times after treatment. Our meta-analysis provided important evidence for the cognitive enhancement of tDCS in patients with MCI and mild-to-moderate AD and discussed its underlying mechanisms.

## 1. Introduction

Alzheimer’s disease (AD) [1] is a common neurodegenerative disease in the elderly and the main cause of dementia; its core symptom is progressive memory loss. With the progression of the disease, patients may also experience aphasia, executive function, and other cognitive impairment, as well as anxiety, depression, irritability, hallucinations, and other neuropsychiatric symptoms [2]. Mild cognitive impairment (MCI) is a state between normal aging and dementia and is considered the preclinical stage of Alzheimer’s disease, with about 5 to 10 percent of patients with mild cognitive impairment possibly developing into dementia each year [3,4,5]. As the population ages, the number of dementia patients is increasing, but drugs, including cholinesterase inhibitors and memantine, cannot effectively improve the cognitive ability of patients [6,7,8]. There is increasing evidence that transcranial direct-current stimulation (tDCS) may be an effective alternative therapy [9].

TDCS is a non-invasive brain stimulation that usually involves placing electrodes on the scalp to apply a weak direct current to modulate cortical function [10,11]. The stimulation of tDCS can be divided into anode stimulation and cathode stimulation; the stimulation current is mostly 1–2 mA, and each stimulation time is tens of minutes, which is considered to be a safe range of stimulation [12,13]. Stimulations of different polarities have diverse effects on the cortex. The anodic tDCS depolarized the resting membrane potential of neurons and increased the excitability of the cortex by increasing the frequency of spontaneous firing of neurons, while the cathodic tDCS hyperpolarized the resting membrane potential of neurons and inhibited the excitability of the cortex by decreasing the firing frequency of neurons [14,15]. However, it has also been observed that, compared with cathodic stimulation of 1 mA, cathodic stimulation of 2 mA can enhance the excitability of the cortex [16]. Changes in cortical excitability caused by tDCS leads to corresponding changes in cortical function and activation [17], that is, changes in synaptic plasticity [18]. Increased cortical excitability and neuroplasticity are considered to be important mechanisms for improving clinical and cognitive abilities in neurodegenerative diseases [19]. In addition, some studies suggest that the cognitive improvement of tDCS may be related to the neural noise produced by TDCS [20,21]. Numerous studies have shown that tDCS can produce varying degrees of therapeutic effects on a variety of neurodegenerative diseases, including Parkinson’s disease, AD, and primary progressive aphasia [22,23,24,25].

Studies have found that tDCS can improve learning and memory disorders in AD-model mice [26,27]. A recent meta-analysis found that tDCS can significantly improve the cognitive function of AD patients, especially when using a low current density [28]. Another meta-analysis found that anodic stimulation of tDCS on DLPFC significantly improved cognitive ability, especially at high-current intensity and density [29]. In terms of the maintenance time of the treatment effect, tDCS can improve the memory impairment of patients with MCI and AD in the short term, but this improvement cannot be maintained for a long time [30]. As a promising treatment for cognitive impairment, it has also been noted that tDCS has no significant therapeutic effect on AD [31,32]. These differences may be related to various factors, such as the stimulus parameters used in various studies, the frequency of stimulation, the means of testing the effect of stimulation, and whether it is combined with cognitive training.

The purpose of this meta-analysis was to evaluate the effect of tDCS on improving the cognitive impairment of patients with MCI and mild-to-moderate AD. Meanwhile, the optimal parameters and duration of the effect were explored.

## 2. Materials and Methods

This study was registered with PROSPERO (CRD42021275672). We followed the stated guidelines for the Preferred Reporting Items for Systematic Reviews and Meta-Analyses (PRISMA) [33]. Two reviewers were independently involved in citation retrieval, study selection, quality assessment, and data extraction. Divergences between reviewers were resolved by consulting the third reviewer.

### 2.1. Search Strategy

In this study, five databases, namely PubMed, Embase, MEDLINE, Web of Science, and The Cochrane Library, were searched; the retrieval time was from the database construction to 31 August 2021. The key words were (“Alzheimer disease” OR “Alzheimer’s disease” OR “AD” OR “mild cognitive impairment” OR “MCI”) AND (“transcranial direct current stimulation” OR “tDCS” OR “direct current stimulation” OR “TES” OR “transcranial stimulation”). The list of references contained in the study was also searched manually to identify any relevant articles.

### 2.2. Study Selection

To ensure the comprehensiveness of the included studies, we accepted both the parallel design trials and the crossover design trials. The inclusion criteria for this study were as follows. (1) Study type: randomized controlled trial (RCT), single-blind, double-blind or non-blind. (2) Subjects: patients with mild-to-moderate AD or MCI who met at least one of the following diagnostic criteria: (a) National Institute of Neurological Communicative Disorders and Stroke/Alzheimer disease and Related Disorders Association(NINCDS/ADRDA), (b) the Diagnostic and Statistical Manual-IV(DSM-IV), (c) Petersen’s criteria, (d) Alzheimer’s disease neuroimaging initiative criteria(ADNI), (e) the Diagnostic and Statistical Manual of Mental Disorders, Fifth Edition(DSM-5), and (f) the criteria of the MCI Working Group of the European Consortium on Alzheimer’s disease. (3) Intervention: The experimental group was treated with tDCS alone (anodic or cathode) or a combination of tDCS and other treatments, and the control group was treated with sham tDCS or a combination of sham tDCS and other treatments. (4) Outcome measures: The primary outcome was the change of cognitive function in MCI and AD patients by tDCS, which was measured by objective cognitive scales. The secondary outcome was the duration of tDCS effect on cognitive impairment.

Exclusion criteria: (1) no RCT; (2) non-English studies; and (3) subjects with vascular dementia, Parkinson’s dementia, Lewy body dementia, frontotemporal dementia, or other types of dementia.

### 2.3. Study Quality Assessment

Methodological quality evaluation was conducted for the included literature, according to the RCT quality evaluation method in Cochrane Systematic Review Manual 5.1.0 [34]. It mainly includes the following six aspects: random sequence generation, allocation concealment, blinding of participants and personnel, blinding of outcome assessment, incomplete outcome data, and selective reporting and other bias. The evaluation results were separated into high risk of bias, low risk of bias, or unclear bias.

### 2.4. Data Extraction and Analysis

In this study, two researchers independently extracted and input data, developed an information extraction table for the literature, and checked each other. Data extraction contents included basic information about the original study, patient characteristics, intervention measures, stimulation site, stimulation times, evaluation methods, and adverse events. If different research protocols are reported in a single study, the data would be included in the meta-analysis as separate units.

The data utilized in this study were the mean difference (MD) and standard deviation (SD) of the changes in the scores of the two groups of patients after treatment and baseline. If the change values were provided in the original study, they were directly extracted; otherwise, the change values were calculated by using formulas [33]. If only images were used to present the results in the original study, the data would be extracted by using Getdata Graph Digitizer (http://getdata-graph-digitizer.com, accessed on 15 October 2021) [28]. When necessary, we contacted the corresponding author to obtain data. If the above methods are not feasible, this study would be excluded.

The formulas are as follows:MD = Mean_final_ − Mean_baseline_;
$$\mathrm{SD}=\sqrt{{\mathrm{SD}}_{\mathrm{baseline}}^{2}+{\mathrm{SD}}_{\mathrm{final}}^{2}-(2\times \mathrm{Corr}\times {\mathrm{SD}}_{\mathrm{baseline}}\times {\mathrm{SD}}_{\mathrm{final}})}$$

All data were analyzed by RevMan 5.4 and STATA 16.0. Weighted mean difference (WMD) was used as the effective value for continuous variables, and 95% confidence interval (CI) was used for interval estimation. I^2^ method was used to determine heterogeneity between studies. When *p* < 0.1 or I^2^ > 50%, the random-effects model would be utilized; otherwise, the fixed-effects model was selected. We searched for possible causes of heterogeneity through meta-regression and sensitivity analysis, and we used subgroup analysis to determine the source of heterogeneity. The leave-one-out cross-validation method was used to test the robustness of the primary outcomes [35]. Publication bias was assessed by observing the asymmetric funnel plot of WMD. Depending on the number of included studies, we chose whether to use Egger’s test for interception [36]. A *p* < 0.05 was considered statistically significant.

## 3. Results

### 3.1. Study Selection and Characteristics

A total of 2480 relevant documents were retrieved in this study. After removing duplicate studies, excluding irrelevant documents, reviews, systematic reviews, animal experiments, etc., 87 documents remained. After browsing through the full text, we excluded non-RCT, inconsistent outcome indicators, and incomplete data. Finally, sixteen works from the literature were included, including seventeen independent studies, in the meta-analysis. The flow diagram of the literature search and selection is shown in Figure 1.

The basic characteristics of the included studies are presented in Table 1. These studies were published between 2008 and 2021.Among the included studies, fifteen studies [37,38,39,40,41,42,43,44,45,46,47,48,49,50] adopted parallel designs, and two studies [17,51] adopted crossover designs. A total of 616 patients were enrolled, with an average age of 72.2 (SD = 7.88). Participants in six studies [39,41,42,47,48,50] were diagnosed with MCI, and participants in the remaining studies [17,37,38,40,43,44,45,46,49,51] were patients with mild-to-moderate AD. Characteristics of tDCS interventions are shown in Table 2. Five studies [38,39,43,46,47] used a combination of anode tDCS and different cognitive training, and two studies [17,44] used both anodic and cathodic tDCS stimulation. One study [17] used a single stimulation, while the others used multiple stimulations. Eight studies [32,38,41,42,43,44,47,49,50] selected the left DLPFC brain region as the stimulation site; one study [48] selected the right DLPFC brain region as the stimulation site; one study [39] selected the left inferior frontal gyrus; and the remaining seven studies selected the temporal-lobe-related brain area as the stimulation site, including (a) temporal cortex bilaterally [51], (b) left temporal lobe [37], (c) temporal areas bilaterally [17,45], (d) left frontotemporal cortex [40], and (e) left lateral temporal cortex [46]. Eight studies [17,37,43,45,46,47,49,51] used a current density of 0.06 mA/cm^2^, one study [42] used a current density of 0.07 mA/cm^2^, five studies [38,41,44,48,50] used a current density of 0.08 mA/cm^2^, one study [39] used a current density of 0.13 mA/cm^2^, and the current density used in two studies [40] was 0.25 mA/cm^2^. All studies evaluated the effect after treatment ends, and ten of them [38,39,41,43,44,46,47,48,49,51] followed up the cognitive function of patients at different time points from one week to six months after the end of treatment.

In terms of cognitive function measurement, different studies used different scales to evaluate the improvement of cognitive function. The detection of overall cognitive function includes Mini-Mental State Examination (MMSE) [37,38,40,41,42,43,44,45,46,51], Alzheimer’s Disease Assessment Scale—Cognitive Subscale (ADAS-Cog) [43,46,49,51], Montreal Cognitive Assessment [45], Cambridge Neuropsychological Test Automated Battery [47,48], Milan Overall Dementia Assessment [40], and Cambridge Cognitive Examination [41]. For the memory domain, the assessment included the word-recall task [41,51], instruction-remembering task [46,51], Rivermead behavioral memory test [38], Rey auditory verbal learning test [38], Tinetti balance scale [38], Tinetti gait scale [38], Word List Memory Test [41], N-back [41], Wechsler Adult Intelligence Scale [41,44], Rey Complex Figure Test [42], Seoul Verbal Learning Test [42], frontal assessment [43], category verbal fluency test [46], digit cancellation task [49], and word-list learning task [49]. The language was assessed by using the Boston Naming Test [41,42], picture-naming task [38], and Battery for Analysis of Aphasic Deficits [38]. Neuropsychiatric Inventory (NPI) [38,46,49] was used to assess the behavioral symptoms, while the word-recognition task [17,49,51] was used to assess recognition memory function. The verbal fluency was assessed by using the Semantic Verbal Fluency test [41], the processing speed was assessed by Symbol Digit Modalities Task [47], and the subjective cognitive function was assessed by Cognitive Failures Questionnaire [47]. Attention was assessed by using the Forward Digit Span Test (FDS) [41,42,46] and Backward Digit Span Test (BDS) [41,42,46]. Visual recognition memory was assessed by using the Visual Recognition Task [51] and Visual Attention Task [17,51]. Verbal memory function was assessed by the California Verbal Learning Test–Second Edition (CVLT-II) [37,39,47]. Trail Making Test part A (TMT-A) [37,38,41,46] measures sustained attention, while Trail Making Test part B (TMT-B) [37,38,41,46], Test of Strategic Learning [39], Delis–Kaplan executive function system [39], Clock Drawing Test [37,41,42,45], Contrasting Program [42], Go-no go Test [42], Controlled Oral Word Association Test [42], and Stroop Test [42] assessed executive function. The Face–Name Association Memory Task [38] was used to assess the patient’s associative memory, and the subjective memory perception was assessed by using Multifactorial Memory Questions [39,50].

### 3.2. Quality Assessment

Among the 17 experiments included in this study, Gangemi et al. [40] conducted two independent experiments. The quality evaluation results of the included literature are shown in Table 3. Most experiments described the random sequence generation method in detail, and seven experiments did not describe the specific situation of allocation concealment. The overall methodological quality of the included experiments was good.

### 3.3. Primary Outcome

Among the 17 experiments included in the study, objective cognitive score scales used in at least three experiments were selected as outcome indicators: MMSE, ADAS-Cog, NPI, word-recognition task, FDS, BDS, CVLT-II, TMT-A, TMT-B, and Clock Drawing Test. Among them, the baseline score of BDS, TMT-A, and TMT-B was not provided in one study [46]; the data of CVLT-II was insufficient in one study [39]; and the version used in TMT-A and TMT-B in one study [41] was different from other studies, so the above scores were excluded. Finally, we analyzed the six scores, namely the MMSE, ADAS-Cog, NPI, FDS, word-recognition task, and Clock Drawing Test. Their forest plots are shown in Figure 2.

### 3.4. MMSE

Eleven experiments [35,36,38,39,40,41,42,43,44,49] were included, with 223 patients in the experimental group and 207 patients in the control group. The heterogeneity test indicated that a random-effects model could be used (I^2^ = 54%, *p* < 0.1). The meta-analysis showed that, compared with the control group, tDCS treatment significantly improved the overall cognitive function assessed by the MMSE of the experimental group, with a combined WMD of 0.99 (95% CI, 0.32 to 1.66; *p* = 0.004; Figure 2a). If studies involving cognitive training were ignored, tDCS significantly improved the overall cognitive function, as assessed by MMSE in the experimental group compared with the control group, with a combined WMD of 1.34 (95% CI, 0.45 to 2.23; *p* = 0.003; Appendix A).

### 3.5. ADAS-Cog

Four experiments [32,46,49,51] were included, with 111 patients in the experimental group and 112 patients in the control group. The heterogeneity test indicated that fixed-effects model could be used (I^2^ = 0%, *p* > 0.1). The meta-analysis showed that tDCS treatment failed to significantly improve overall cognitive function, as assessed by ADAS-Cog in the experimental group, with a combined WMD of −0.46 (95% CI, −1.43 to 0.51; *p* = 0.35; Figure 2b).

### 3.6. NPI

Three experiments [38,46,49] were included, with 101 patients in the experimental group and 96 patients in the control group. The heterogeneity test indicated that the fixed-effects model could be used (I^2^ = 0%, *p* > 0.1). The meta-analysis showed that the tDCS treatment failed to significantly improve behavioral symptoms in the treatment group compared with the control group, with a combined WMD of 1.00 (95% CI, −0.02 to 2.03; *p* = 0.05; Figure 2c).

### 3.7. Word-Recognition Task

Three experiments [17,49,51] were included, with 58 patients in the experimental group and 47 patients in the control group. The heterogeneity test indicated that the random-effects model could be used (I^2^ = 64%, *p* < 0.1). The meta-analysis showed that the tDCS treatment did not significantly improve recognition memory function in the experimental group compared with the control group, with a combined WMD of 0.53 (95% CI, −0.52 to 1.58; *p* = 0.32; Figure 2d).

### 3.8. FDS

Three experiments [41,42,46] were included, with 109 patients in the experimental group and 100 patients in the control group. The heterogeneity test indicated that the fixed-effects model could be used (I^2^ = 0%, *p* > 0.1). The meta-analysis showed that the tDCS treatment did not significantly improve attention in the treatment group compared with the control group, with a combined WMD of 0.01 (95% CI, −0.17 to 0.20; *p* = 0.89; Figure 2e).

### 3.9. Clock Drawing Test

Four experiments [37,41,42,45] were included, with 75 patients in the experimental group and 70 patients in the control group. The heterogeneity test showed that the random-effects model could be used (I^2^ = 82%, *p* < 0.1). The meta-analysis showed that tDCS treatment did not significantly improve the executive function in the treatment group compared with the control group, with a combined WMD of −0.21 (95% CI, −0.86 to 0.45; *p* = 0.54; Figure 2f).

### 3.10. Subgroup Analysis and Meta-Regression

A subgroup analysis was used to identify variables that might affect the heterogeneity for the MMSE score (Figure 3). In view of the complexity of tDCS parameters in each study, the factors most likely to affect heterogeneity were selected for subgroup analysis, including stimulate sites, number of sessions, current density, and disease level. Among them, the number of sessions was comprehensively analyzed according to the number of tDCS treatments and the days required to complete the treatment, and the current density was divided according to the included study conditions. The results showed that the stimulation of the left DLPFC (WMD = 0.37; 95% CI, −0.15 to 0.90; *p* = 0.16) did not significantly improve the MMSE score, and stimulated temporal-lobe-related brain areas (WMD = 1.68; 95% CI, 0.41 to 2.95; *p* = 0.009) can significantly improve the MMSE score. The subgroup analysis of the number of stimulations showed that the number of stimulations that ranged from 5 to 10 (WMD = 0.61; 95% CI, −0.11 to 1.32; *p* = 0.10) had no improvement in cognitive function; meanwhile, the number of stimulations that ranged from 10 to 15 (WMD = 0.84; 95% CI, 0.04 to 1.64; *p* = 0.04) improved cognitive function, and the number of stimulations that was greater than or equal to 15 times (WMD = 3.52; 95% CI, 1.51 to 5.53; *p* = 0.0006) significantly improved cognitive function. The subgroup analysis of current density showed that the current density was 0.06 mA/cm^2^ (WMD = 1.04; 95% CI, −0.05 to 2.13; *p* = 0.06), 0.07 mA/cm^2^ (WMD = 2.60; 95% CI, −1.56 to 6.76; *p* = 0.22), or 0.08 mA/cm^2^ (WMD = 0.34; 95% CI, −0.20 to 0.87; *p* = 0.22) and did not improve cognitive function, while 2.5 mA/cm^2^ (WMD = 2.84; 95% CI, 1.16 to 4.51; *p* = 0.0009) significantly improved cognitive function. Compared with MCI patients (WMD = 0.54; 95% CI, −1.07 to 2.14; *p* = 0.51), AD patients benefited more in regard to cognitive function after tDCS stimulation (WMD = 1.10; 95% CI, 0.32 to 1.87; *p* = 0.005).

A subgroup meta-regression analysis (Appendix A) was conducted to explore the influence of different study characteristics on the MMSE score, and the overall results were relatively robust.

### 3.11. Sensitivity Analysis and Publication Bias

The heterogeneity of the MMSE score included in the study is relatively high (I^2^ = 54%), so a sensitivity analysis was performed (Figure 4). When we ignored any of the studies, the overall results were not significantly different. Therefore, the results remain stable and robust. There was no suggestion of a small study effect based on visual inspection of the funnel plot (Figure 5). The results of the Egger’s test (*p* = 0.144) and Begg’s test (*p* = 0.350) showed that there was no potential publication bias (Appendix A).

### 3.12. Secondary Results

Effect at different time points after tDCS stimulation:

To evaluate the effects of tDCS on MCI or AD patients at different time points after treatment, we divided the MMSE score into three subgroups (≤1 month, ≤2 months, and >2 months) according to different follow-up times. The forest plot (Figure 6) shows that the total effective value at different follow-up times after the end of treatment was 0.95 (95% CI, −0.42 to 2.33; *p* = 0.17). Among them, the WMD of the ≤1 month group was 0.91 (95% CI, −1.09 to 2.92; *p* = 0.37), the WMD of the ≤2 months group was 2.01 (95% CI, −1.55 to 5.57; *p* = 0.27), and the >2 months group WMD was −0.42 (95% CI, −2.01 to 1.18; *p* = 0.61).

## 4. Discussion

This study systematically evaluated the effect of tDCS stimulation on the cognitive function of patients with MCI and mild-to-moderate AD and the effect at different time points after treatment, including 16 studies. The results showed that tDCS significantly improved the overall cognitive function of patients with MCI and mild-to-moderate AD evaluated by MMSE, but it had no significant improvement on the ADAS-Cog score. Meanwhile, tDCS treatment failed to significantly improve the behavioral symptoms, recognition memory function, attention, and executive function of patients with MCI and mild-to-moderate AD. The results of the subgroup analysis showed that the stimulation of temporal-lobe-related brain regions, the number of stimulations ≥10, and the current density of 2.5 mA/cm^2^ were better than for the stimulation of left DLPFC; the number of stimulations was between 5 and 10; the current density was 0.06, 0.07, or 0.08 mA/cm^2^. Moreover, compared with MCI patients, patients with mild-to-moderate AD might benefit more from tDCS treatment. After the end of the tDCS treatment, there were no statistically significant differences in MMSE score changes at the follow-ups after 1 month, 2 months, and more than 2 months.

After receiving tDCS treatment, the overall cognitive function of patients with MCI and mild-to-moderate AD evaluated by MMSE had been significantly improved, which was similar to the results of a previous meta-analysis [28]. The MMSE score is the most commonly used screening tool to measure cognitive impairment in clinical practice [52]. The research selection, data extraction, and bias-risk assessment were conducted by an independent reviewer in this study, and this helped us avoid the selective reporting of specific results and further strengthened the effectiveness of our meta-analysis. The ADAS-Cog score did not change significantly after tDCS treatment in this study; this outcome might be linked to the small number of trials included in the score. In addition to the impairment of overall cognitive function, AD patients also have defects in working memory, executive function, attention, language fluency, etc. [53,54]. Currently, studies have reported that tDCS has a significant therapeutic effect on speech–motor learning [55], seizure-related attention deficit [56], executive dysfunction [57], etc. However, we did not find a significant therapeutic effect with MCI and mild-to-moderate AD.

The effects of tDCS stimulation in different brain regions might be different. Our results revealed that the stimulation of the temporal-lobe-related brain regions had better cognitive improvement than the left DLPFC stimulation. Numerous studies have shown that the temporal lobe is related to short-term and long-term memory storage [58,59], and the DLPFC is not only involved in cortical and subcortical functional connectivity, but also plays an important role in maintaining executive memory and cognition and working memory [60,61]. Therefore, non-invasive transcranial stimulation often uses these two brain regions as the stimulation site. It has been found that transcranial random noise stimulation in the lateral temporal lobe can significantly improve epileptic-related memory deficits [62]. This may mean that tDCS can also improve cognitive function by producing neural noise in the temporal lobe. Transcranial magnetic stimulation (rTMS), another non-invasive brain stimulation modality that works by altering cortical excitability, was considered to be effective in cognitive improvement by stimulating the right DLPFC in a meta-study [63]. All the experiments that stimulate DLPFC included in this study used the left side, and this might be the reason for its insignificant effect. In addition to the two stimulation sites mentioned above, this study also included one experiment stimulating right DLPFC and one stimulating the inferior frontal gyrus (IFG). The cognitive evaluation scale of the two experiments was less than three experiments, so no meta-analysis was conducted. Increasing the exploration of brain regions where tDCS might be effective and unifying the evaluation criteria for cognitive function might give us more insight into the effect of tDCS on cognitive improvement.

Multiple studies have shown that tDCS treatment has a cumulative effect [64,65,66] and repeated tDCS treatment may be more effective than a single treatment. In this study, we found no significant improvement in MMSE scores when the total number of stimulations was between 5 and 10. When the number of stimulations was between 10 and 15, the MMSE score increased significantly. When the number of stimulations was more than 15, the MMSE score improved more significantly. Studies have also shown that tDCS stimulation at different time intervals may affect its therapeutic effect. The cumulative effect was obvious when tDCS was applied continuously, but there was no obvious cumulative effect when tDCS was applied every 2 days or weekly [67,68]. This may be related to the current state of the brain, as studies have found that fewer active neurons are more likely to be promoted by subsequent external stimulation (tDCS) [69]. Exploring the underlying mechanism of the cumulative effect of tDCS and finding the best time to produce the cumulative effect will be an important research direction for tDCS to improve the cognitive function of patients with MCI and mild-to-moderate AD.

The current density is the ratio of the current to the size of Montage, which is an important parameter in tDCS treatment. In our study, the current densities of 0.06, 0.07, and 0.08 mA/cm^2^ could not improve the MMSE score, but 2.5 mA/cm^2^ could produce significant cognitive improvement. Among them, 0.06 and 0.08 mA/cm^2^ are commonly used parameters in clinical studies. As is well-known, the effect induced by 0.08 mA/cm^2^ is generally greater than the effect induced by 0.06 mA/cm^2^ [70], which is slightly different from our conclusions. It has been found that tDCS in MCI patients may produce smaller current density in the brain regions targeted by tDCS than in healthy elderly people when given the same current density of tDCS, due to greater brain atrophy [71]. Therefore, it is possible that only a relatively large current density can produce significant effects on tDCS treatment. In terms of the selection of Montage size, Foerster et al. found that the stimulation of a small Montage size was more specific than that of a large Montage size at the same current density [72]. In this study, when the current density was 2.5 mA/cm^2^, the size of Montage was only 0.8 cm^2^, which might be the reason for the significant improvement in cognitive function. However, 2.5 mA/cm^2^ was the first time to be applied to patients with AD; although that study included two independent experiments, the small number of subjects included might reduce the credibility of the conclusions. Therefore, more studies with different current densities are required to determine the best parameters for tDCS to improve the cognitive function of patients with MCI and mild-to-moderate AD.

Among patients at different disease degrees, tDCS significantly improved the cognitive function of AD patients but not MCI patients. A recent meta-analysis [73] found that high-frequency rTMS improved cognitive function in patients with AD, but had no significant effect on cognitive function in patients with MCI. The cognitive improvement of tDCS was also greater only in patients with AD than in patients with MCI [73,74]. This might be related to the ceiling effect of MMSE [75]; that is, it is difficult for MCI patients to detect large changes in MMSE scores. However, there were only two experiments of MCI patients included in this study, and we need to interpret the results more cautiously.

How long the cognitive benefit would be of tDCS in patients of MCI and AD is another important issue that people care about. In the current meta-analysis, we did not find that the improvement effect of tDCS on cognitive function could be maintained for a long time. In addition to the acute effects on brain functions, specific tDCS protocols have been reported to induce long-lasting alterations of cortical excitability and activity [18]. Therefore, tDCS treatment has the potential for lasting benefits. Given the small number of trials with follow-up records included in this study, the current results need to be interpreted with caution, and more studies are needed to provide evidence for the long-term effects of tDCS.

## 5. Limitations

This meta-analysis also has certain limitations. On the one hand, constrained by the inclusion criteria and the different evaluation criteria between the studies, the final sample size for meta-analysis was small; thus, it might have some limitations on the statistics of tDCS treatment effects. On the other hand, this study also included the experiment of tDCS alone and the experiment of combining tDCS with other cognitive training. The cognitive training methods of each study are different and may have a certain impact on the results. In terms of heterogeneity test, there was no subgroup analysis of treatment time, follow-up time, race, blindness of study design, and other factors, due to the lack of parameters, thus resulting in insufficient detailed parameters for the tDCS treatment.

## 6. Conclusions

In summary, the existing evidence showed that tDCS can significantly improve the overall cognitive function of patients with MCI and mild-to-moderate AD, especially in the stimulation of temporal-lobe-related brain regions; the number of stimulations was greater than or equal to 10 times, and the current density is 2.5 mA/cm^2^. Meanwhile, AD patients might benefit more than MCI patients. In terms of behavioral symptoms, recognition memory function, attention, and executive function, tDCS treatment did not bring significant benefits. Finally, the therapeutic effect of tDCS was only obvious at the end of the tDCS, and no effect of tDCS on improving cognitive function was found in the follow-up of 1 month, 2 months, and more than 2 months.

## Figures and Tables

**Figure 1 brainsci-12-00562-f001:**
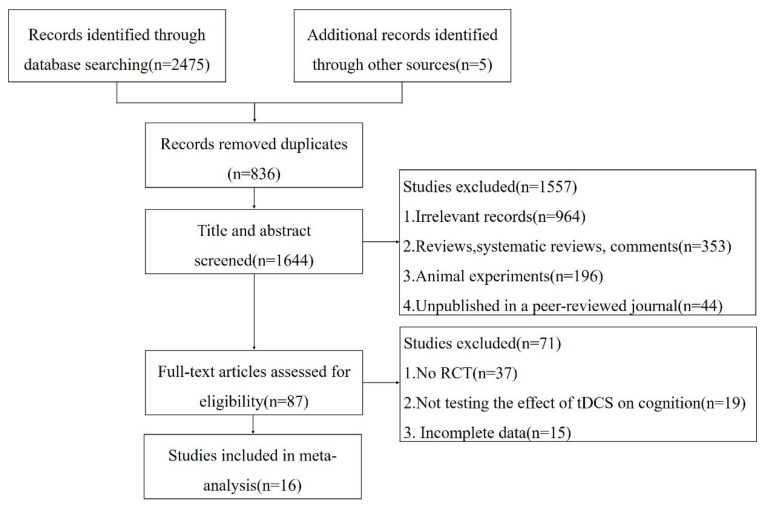
Flow diagram of the screening process of the literature.

**Figure 2 brainsci-12-00562-f002:**
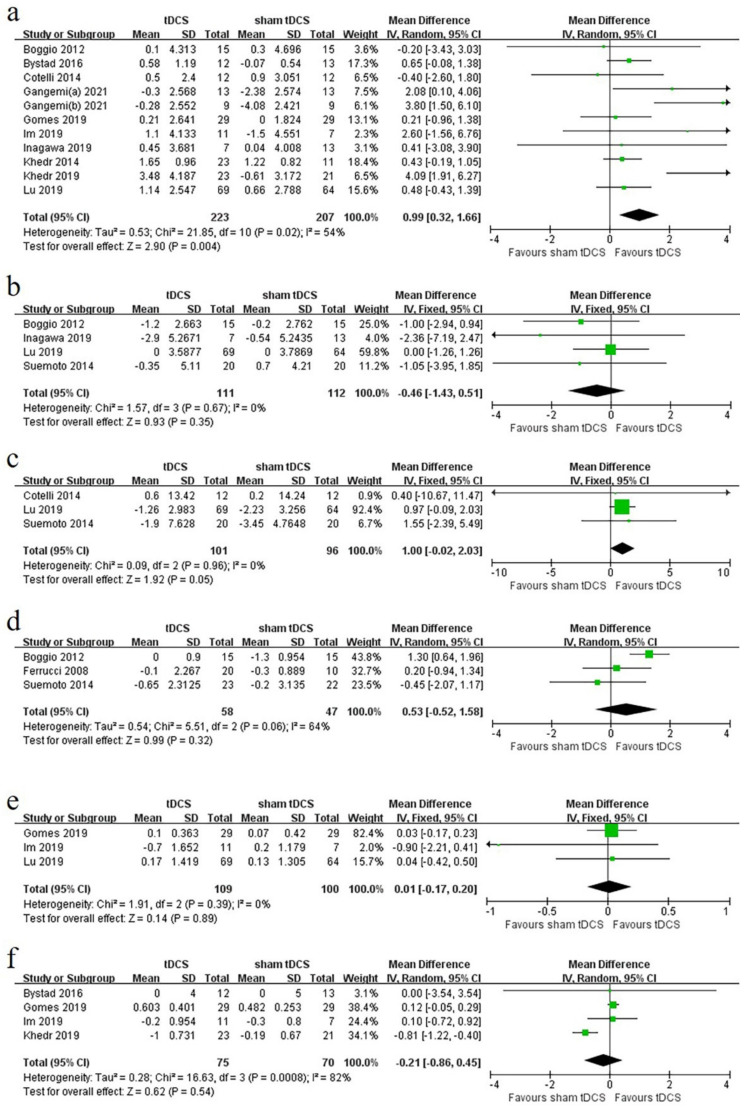
Forest plots of mean change from baseline, based on different scales: (**a**) Mini-Mental State Examination (MMSE), the order of references is [51,37,38,40,40,41,42,44,45,46]. (**b**) Alzheimer’s Disease Assessment Scale–Cognitive Subscale (ADAS-Cog), the order of references is [51,43,46,49]. (**c**) Neuropsychiatric Inventory (NPI), the order of references is [38,46,49]. (**d**) word-recognition task, the order of references is [51,17,49]. (**e**) Forward Digital Span (FDS), the order of references is [41,42,46]. (**f**) Clock Drawing Test, the order of references is [37,41,42,45]. Independent studies in the same literature are distinguished by (**a**,**b**). Error bars are 95% confidential intervals.

**Figure 3 brainsci-12-00562-f003:**
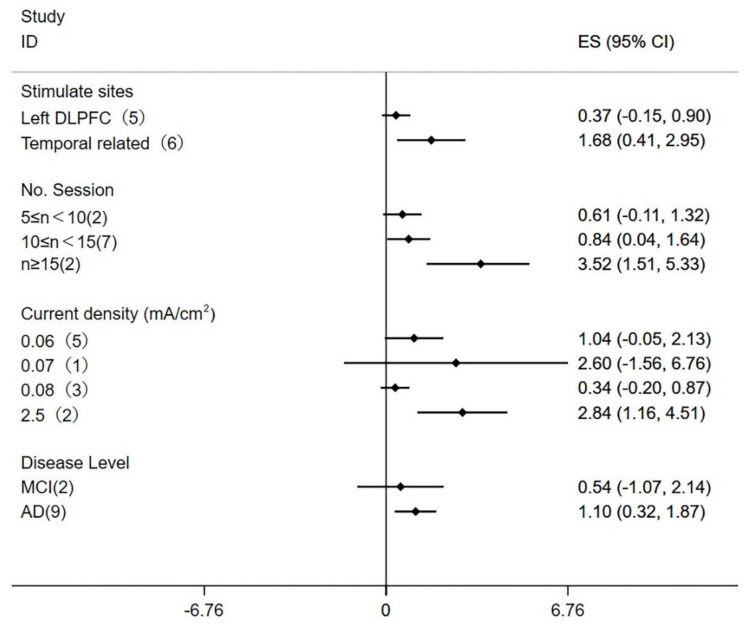
Subgroup analyses of MMSE.

**Figure 4 brainsci-12-00562-f004:**
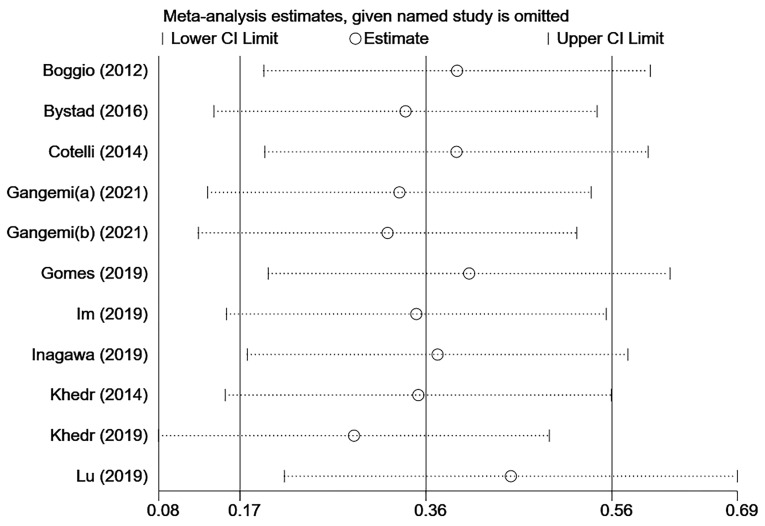
Sensitivity analyses of MMSE. The order of references is [51,37,38,40,40,41,42,43,44,45,46].

**Figure 5 brainsci-12-00562-f005:**
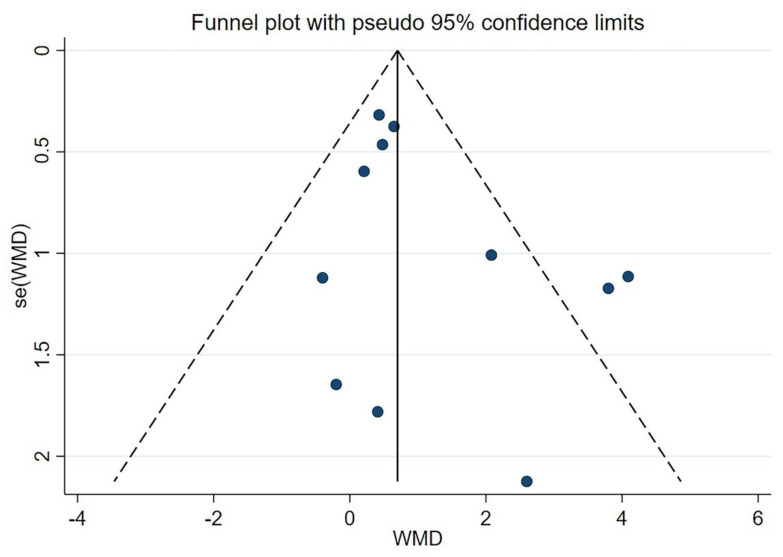
Funnel plot of MMSE.

**Figure 6 brainsci-12-00562-f006:**
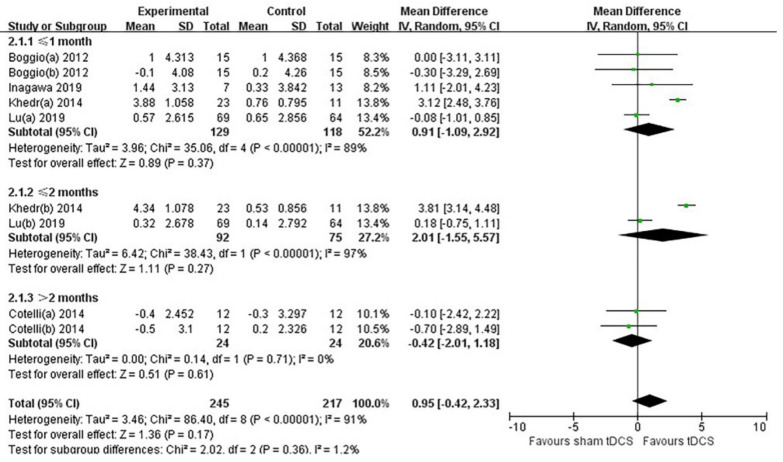
Forest plots of the effects of tDCS on MCI or mild-to-moderate AD patients at different time points after treatment. The order of references is [51,37,38,40,40,41,42,43,44,45,46]. The results of different follow-up times in the same study are represented by (a) and (b).

**Table 1 brainsci-12-00562-t001:** The basic characteristics of the included studies.

Study (Time)	Sample Size	Design	Diagnosis	Gender (M/F)	Age (y)	Education (y)	Duration of Disease(y)	Outcomes for Cognition Function
Boggio et al. (2012) [51]	NE:15 NC:15	Crossover	AD	8/7	78.95 ± 8.07	14.42 ± 3.65	4.39 ± 1.88	MMSE, VAT, ADAS-Cog, Word recall, Word recognition, Instruction remembering, VRT
Bystad et al. (2016) [37]	NE:12 NC:13	Parallel	AD	7/5 7/6	70.0 ± 8.0 75.0 ± 8.7	NR	NR	CVLT-II, MMSE, Clock-drawing test, TMT-A, TMT-B
Cotelli et al. (2014) [38]	NE:12 NC:12	Parallel	AD	2/10 3/9	76.6 ± 4.6 74.7 ± 6.1	5.5 ± 2.4 8.9 ± 5.1	NR	FNAT, MMSE, Tinetti balance scale, Tinetti gait scale, NPI, Picture naming task, BADA, Rivermead behavioral memory test, Rey auditory verbal learning test, TMT-A, TMT-B
Das et al. (2019) [39]	NE:12 NC:10	Parallel	MCI	8/4 8/2	62.58 ± 8.43 63.30 ± 7.38	17.92 ± 3.94 16.20 ± 1.75	NR	TOSL, DKEFS, CVLT, MMQ
Ferrucci et al. (2008) [17]	NEa:10 NEb:10 NC:10	Crossover	AD	3/7	75.2 ± 7.3	10.9 ± 4.8	NR	Word recognition task, VAT
Gangemi(a) et al. (2021) [40]	NE:13 NC:13	Parallel	AD	NR	67.5 ± 2.8 69.01 ± 3.1	6.5 ± 2.0 6.1 ± 2.1	NR	MMSE, MODA
Gangemi(b) et al. (2021) [40]	NE:9 NC:9	Parallel	AD	NR	68.5 ± 2.8 68.7 ± 3.1	6.7 ± 2.0 6.2 ± 2.7	NR	MMSE, MODA
Gomes et al. (2019) [41]	NE:29 NC:29	Parallel	MCI	9/20 7/22	73.0 ± 9.2 71.6 ± 7.9	NR	NR	CAMCOG, MMSE, TMT-A, TMT-B, SVF, BNT, Clock-drawing test, WLMT, WAIS, N-back, FDS, BDS
Im et al. (2019) [42]	NE:11 NC:7	Parallel	MCI	1/10 2/5	71.9 ± 9.2 74.9 ± 5.0	6.3 ± 3.8 5.4 ± 5.9	NR	MMSE, FDS, BDS, BNT, SVLT, COWAT, RCFT, Contrasting Program, Go-no go Test, Stroop Test, Clock-drawing test
Inagawa et al. (2019) [43]	NE:7 NC:13	Parallel	AD	3/4 7/6	76.6 ± 5.7 76.2 ± 7.7	NR	0.9 ± 1.2 1.2 ± 1.5	ADAS-Cog, MMSE, FAB
Khedr et al. (2014) [44]	NEa:11 NEb:12 NC:11	Parallel	AD	6/5 8/4 5/6	68.5 ± 7.2 70.7 ± 5.4 67.3 ± 5.9	NR	3.0 ± 2.6 2.9 ± 1.9 3.5 ± 1.7	MMSE, WAIS
Khedr et al. (2019) [45]	NE:23 NC:21	Parallel	AD	13/10 13/8	64.22 ± 3.64 65.23 ± 4.52	1.17 ± 0.48 1.17 ± 0.39	4.04 ± 2.83 3.52 ± 1.96	MMSE, Clock-drawing test, MoCA
Lu et al. (2019) [46]	NE:69 NC:64	Parallel	AD	21/42 17/36	74.2 ± 6.7 74.5 ± 6.6	7.3 ± 4.8 6.5 ± 4.3	NR	ADAS-Cog, MMSE, NPI, CVFT, FDS, BDS, TMT-A, TMT-B
Martin et al. (2019) [47]	NE:33 NC:35	Parallel	MCI	13/20 10/25	71.8 ± 6.39 71.6 ± 6.35	14.5 ± 3.51 14.9 ± 3.23	NR	CVLT-II, CANTAB, SDMT, CFQ
Stonsaovapak et al. (2020) [48]	NE:23 NC:22	Parallel	MCI	2/21 2/20	68.39 ± 8.37 69.68 ± 7.60	NR	NR	CANTAB
Suemoto et al. (2014) [49]	NE:20 NC:20	Parallel	AD	5/15 7/13	79.4 ± 7.1 81.6 ± 8.0	5 ± 4.2 4.5 ± 3.9	NR	NPI, ADAS-Cog, Digit cancellation task, Word list learning task, Word recognition task
Yun et al. (2016) [50]	NE:8 NC:8	Parallel	MCI	3/5 2/6	74.75 ± 7.47 73.12 ± 4.25	8.06 ± 4.93 5.56 ± 2.41	NR	MMQ

Data are expressed as mean ± SD. Independent studies in the same literature are distinguished by (a) and (b). AD, Alzheimer disease; MCI, mild cognitive impairment; M, male; F, female; NR, not reported; MMSE, Mini-Mental State Examination; VAT, Visual Attention Task; ADAS-Cog, Alzheimer’s Disease Assessment Scale—Cognitive Subscale; VRT, Visual Recognition Task; CVLT-II, California Verbal Learning Test—Second Edition; TMT-A, Trail Making Test parts A; TMT-B, Trail Making Test parts B; FNAT, Face–Name Association Memory Task; NPI, Neuropsychiatric Inventory; BADA, Battery for Analysis of Aphasic Deficits; TOSL, Test of Strategic Learning; DKEFS, Delis–Kaplan executive function system; MMQ, Multifactorial Memory Questions; MODA, Milan Overall Dementia Assessment; CAMCOG, Cambridge Cognitive Examination; SVF, Semantic Verbal Fluency test; BNT, Boston Naming Test; SVLT, Seoul Verbal Learning Test; COWAT, Controlled Oral Word Association Test; RCFT, Rey Complex Figure Test; FAB, frontal assessment battery; WAIS, Wechsler Adult Intelligence Scale; MoCA, Montreal Cognitive Assessment; FDS, Forward Digit Span Test; BDS, Backward Digit Span Test; CVFT, category verbal fluency test; CANTAB, Cambridge Neuropsychological Test Automated Battery; SDMT, Symbol Digit Modalities Task; CFQ, Cognitive Failures.

**Table 2 brainsci-12-00562-t002:** The characteristics of tDCS interventions.

Study (Time)	Type of Stimulation	Number of Sessions	Duration (min)	Stimulation Site	Current (mA)	Montage Size (cm^2^)	Stimulation Model	Adverse Effects
Boggio et al. (2012) [51]	Anode Sham	per day for 5 consecutive days	30	Temporal cortex bilaterally	2	35	Offline	No adverse effects were recorded after five daily tDCS sessions
Bystad et al. (2016) [37]	Anode Sham	6 sessions for 10 days	30	Left temporal lobe	2	35	Offline	No adverse effects were reported
Cotelli et al. (2014) [38]	Anode+ICMT Sham+ICMT	5 sessions per week for 2 weeks	25	Left DLPFC	2	25	Online	NR
Das et al. (2019) [39]	Anode+SMART Sham+SMART	8 sessions for 4 weeks	20	Left IFG	2	15	Offline	NR
Ferrucci et al. (2008) [17]	Anodal Cathodal Sham	1 session	15	Temporoparietal areas bilaterally	1.5	25	Offline	NR
Gangemi(a) et al. (2021) [40]	Anode Sham	Daily, for 10 days	20	Left frontotemporal cortex	2	0.8	Offline	NR
Gangemi(b) et al. (2021) [40]	Anode Sham	10 sessions each month for 8 months	20	Left frontotemporal cortex	2	0.8	Offline	NR
Gomes et al. (2019) [41]	Anode Sham	Twice per week for 5 weeks	30	Left DLPFC	2	25	Offline	NR
Im et al. (2019) [42]	Anode Sham	Daily, for 6 months	30	Left DLPFC	2	28	Offline	NR
Inagawa et al. (2019) [43]	Anode+CT| Sham+CT	2 sessions per day for 5 consecutive days	20	Left DLPFC	2	35	Online	Neither severe adverse events nor the need for medications caused by adverse events
Khedr et al. (2014) - [44]	Anodal Cathodal Sham	Daily, for 10 days	25	Left DLPFC	2	24	Offline	Two patients under active stimulation recorded itching, headache, and dizziness that were disappear after few hours
Khedr et al. (2019) [45]	Anode Sham	5 sessions per week for 2 consecutive weeks	20 (each side)	Left TP lobe and right TP lobe	2	35	Offline	All the patients tolerated tDCS well without major adverse effects
Lu et al. (2019) [46]	Anode+WMT Sham+WMT	3 sessions per week for 4 weeks	20	Left LTC	2	35	Offline	three cases had skin lesions under the cathodal electrode during the repeated sessions of tDCS
Martin et al. (2019) [47]	Anode+CT Sham+CT	3 sessions per week for 5 weeks	30	Left DLPFC	2	35	Online	No adverse effects were reported
Stonsaovapak et al. (2020) [48]	Anode Sham	3 times per week for 4 weeks	20	Right DLPFC	2	25	Offline	Dizziness was found in one participant from the atDCS group. All side effects disappeared within 24 hours
Suemoto et al. (2014) [49]	Anode Sham	3 sessions per week for 2 weeks	20	Left DLPFC	2	35	Offline	TDCS was well tolerated and not associated with significant adverse effects
Yun et al. (2016) [50]	Anode Sham	3 sessions per week for 3 weeks	30	Left DLPFC	2	25	Offline	No patient reported adverse effects

NR, not reported; CT, cognitive training; ICMT, individualized computerized memory training; SMART, strategic memory and advanced reasoning training; WMT, working-memory training; DLPFC, dorsolateral prefrontal cortex; IFG, inferior frontal gyrus; LTC, lateral temporal cortex; TP, temporoparietal.

**Table 3 brainsci-12-00562-t003:** Assessment of risk of bias for included studies.

Study	Sequence Generation	Allocation Concealment	Blinding of Participants	Personnel and Outcomes Assessors	Incomplete Outcome Data	Selective Outcomes Reporting	Baseline Characteristics
Boggio (2012) [51]	?	?	?	+	+	?	-
Bystad (2016) [37]	+	+	?	?	+	+	?
Cotelli (2014) [38]	?	?	+	+	?	+	+
Das (2019) [39]	+	+	+	+	?	+	?
Ferrucci (2008) [17]	+	+	+	+	+	?	?
Gangemi(a) (2021) [40]	?	?	+	+	?	+	?
Gangemi(b) (2021) [40]	?	?	+	+	?	+	?
Gomes (2019) [41]	?	?	?	+	?	+	-
Im (2019) [42]	+	+	?	+	+	+	+
Inagawa (2019) [43]	+	+	?	+	+	+	+
Khedr (2014) [44]	+	?	+	+	+	+	?
Khedr (2019) [45]	+	?	+	+	+	+	?
Lu (2019) [46]	+	?	+	+	+	+	+
Martin (2019) [47]	+	+	?	+	?	+	+
Stonsaovapak (2020) [48]	+	+	+	+	+	+	+
Suemoto (2014) [49]	+	+	+	+	+	+	+
Yun (2016) [50]	+	+	+	+	+	+	?

Note: + low, - high, and ? uncertain.

## Data Availability

https://doi.org/10.5061/dryad.dr7sqvb1j (accessed on 18 April 2022).

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
