# Peer review of "Transcranial Direct Current Stimulation Enhances Cognitive Function in Patients with Mild Cognitive Impairment and Early/Mid Alzheimer’s Disease: A Systematic Review and Meta-Analysis"

_brainsci, 2022, doi:10.3390/brainsci12050562_

Round 1

Reviewer 1 Report

Overview:

The paper addresses the efficacy and parameters of 2mA cathodic tDCS stimulation across the cortex on AD and MCI patients. The paper notes improved overall cognitive functioning (although only improvements on MMSE scores were noted). No significant changes in individual cognitive domains of attention, memory and executive functioning was observed, and no long lasting effects of the treatment.

Major issues:

The main effect of improved MMSE scores after the intervention is not supported by the forest plot shown in Figure 2 (a). The results actually show a preference towards the sham condition in this case. This would have a significant bearing on the discussion and overall recommendations for using tDCS in MCI and AD patients.

There is no clear description of the sham conditions used within the studies. This maybe important if different shams were used, and the relative effectiveness of the sham within studies. This can also influence interpretation of the data.

Minor corrections:

Page 2, lines 49-53: This description is in the past tense when in theory given the context of the preceding and post sentences it should be made in the present tense i.e. “The anodic tDCS depolarises the resting membrane potential and increases the …”

Page 2, line 55: … caused by tDCS lead to … should state “leads to”

Page 2, Line 60: should read “… Parkinson’s Disease (PD) and primary progressive aphasia”…

Page 2, line 65: Can tDCS really pin point stimulation within only DLPFC in this study given its broad stimulation effects.

Page 2, line 81 (and 117): Statement here needed on risk of bias (i.e. quality assessment) and if this was also performed by both reviewers.

Page 2, line 89: were articles also identified during the search based on reading the papers and identifying within those papers relevant articles?

Page 3. Line 108: Should read “(3) Subjects with vascular dementia …”

Page 3, line 128: Can you be more specific about the formula used here.

Page 4, line 149-150: The flow diagram does not reveal the 16 literatures, and 17 independent studies used and states the final outcome in the meta-analysis is 14? Clarity needed here.

Page 4, line 155-156: Here the number of studies included is 17, not 14 as stated in the flow diagram?

Page 4, line 157: Age range and standard deviation would be good to include alongside the mean age.

Page 4, line 157: 14 studies noted here which matches the flow diagram.

Page 4, line 163 = what is meant by “multiple stimulation” here?

Page 4, lines 163-169: 17 students are noted here in stimulation sites which doesn’t match the original 14?

Page 4, lines 169-173: Again 17 studies where parameters are provided.

Page 4, line 173: “ten of them […]” but only 8 cited? Clarity needed.

Page 6: Table 2 resolution and font size is very small making it difficult to read.

Page 8: Labelling on the table of the tests (a-f) would help clarity.

Page 9, line 251-256: The meta analysis for the MMSE seems to favour the sham condition? Given the position of the diamond. Can you elaborate on why a significant effect of the intervention is reported here?

Page 9, line 263:  The NPI seem to reveal a borderline significance to the sham condition with confidence intervals showing just a very small overlap. Is this worth discussing in your results?

Page 10, line 290: Might be worth mentioning in the text how many studies were included in this sub-group analysis of Left DLPFC and temporal related areas. In Figure 3 the numbers in brackets are not specified.

Page 11, line 316: “When we ignored any of the studies, the overall results were not significantly different.” I do not understand this sentence. Can you clarify what studies you ignored here?

Page 11, Figure 4: Can you provide an X axis label.

Page 11, Figure 5: In the legend please state in full what WMD stands for.

Page 12, line 341: Again there is some confusion here on number of studies included. Was it 16 articles with one article including 2 studies and hence 17 in total ?

Page 12, line 341-343: Given the point made about the MMSE favouring the Sham condition. This interpretation seems unfounded?

Page 13, line 365: Is seizor the correct use of the word here?

Page 13, line 371-374: Caution is needed here given the depth of stimulation for tDCS and the claim that temporal lobe tDCS induces affects in the Human hippocampus. According to my knowledge this has only been demonstrated in rat/mice slices and in vivo rat preparations and translation into the much large human brain needs caution. Evidence for this effect in humans is needed here.

Page 13, line 401: Some discussion on the links between higher current densities/montage size and potential adverse effects is needed here. Given Table 2 seems to reveal more adverse effects on participants with larger montages?

Page13, line 410-411: Can the authors provide any supporting evidence to why effects might be stronger in AD patients?

Page 14, line 434: Given only MMSE revealed significant effects, caution is needed when stating that tDCS can significantly improve overall cognitive function. What about potential direct learning effects of the MMSE within these studies (see Lee, Lin & Chiu, 2020)? There is a reported higher practise effect than other cognitive screening tools in individuals with dementia with an effect size of around 0.26 (Cohen’s d).

Author Response

Thank you very much for your valuable advice, which is very helpful to our manuscript. We have replied each opinion you mentioned one by one. Please find the specific content in the attachment.

Reviewer 2 Report

 In this manuscript, the authors conducted a systematic review and meta-analysis on tDCS effects on improving cognitive functions in both patients with mild cognitive impairment (MCI) and early-mid Alzheimer’s Disease (AD). They found active tDCS treatment significantly improved cognitive functions indexed by the Mini-Mental State Examination (MMSE) for patients with AD and MCI. Subgroup analysis indicated that the treatment would be more efficacious when stimulating temporal lobe-related brain areas, number of sessions over 10, and current density at 2.5 mA/cm2.

Considering patients with MCI may develop into AD, it may provide additional information to the field to combine the MCI sample and AD sample to evaluate the tDCS treatment effects on cognitive functions in those patients. Besides the main findings from MMSE demonstrating the effectiveness of tDCS on improving cognitive functions, the findings from the subgroup analysis were also informative in terms of the proper parameters for tDCS to generate larger effects. Although there are recent meta-analysis papers on similar topics (Cai et al., 2019, Alzheimer Dis Assoc Disord; Wang et al., 2021, Alzheimer Dis Assoc Disord), the current study provides essential information and may be of clinical importance. I have a few minor comments that I think needs to be addressed.

1, From the Tables and Figures, it seems clear that there were 17 studies involved in the meta-analysis. However, there are several occasions throughout the text that different numbers of studies involved were mentioned. For example, Page 4 line 149-150, ‘sixteen literatures were included, including seventeen independent studies, into the meta-analysis.’, bottom box in Figure 1 where n=14 was mentioned, page 12 line 341, ’16 studies’ was stated.

2, Please double check whether the labels for ‘Favors tDCS’ and ‘Favors sham tDCS’ were correct in the Figures. For example, Figure 2a showed that the overall effects favored sham tDCS.

3, There are also some typo and/or editing issues:

3.1, On page 2, line 78. ‘The’ needs to be removed before ‘This study…’;

3.2, there are 11 studies with AD, but only 10 citations were given at page 4 line 159;

3.3, in figure 1, number of studies excluded (n=64) does not match the sum of its sub-categories (i.e., 37+19+7=63); also in Figure 1, the number of eligible studies (n=87) minus the number of excluded studies (n=64) does not match the final number of studies included for meta-analysis (n=17);  

3.4, mA/cm2 should be mA/cm2

3.5, All references in the Reference Section were double-numbered.

Author Response

(The authors gave the same response as above.)

Reviewer 3 Report

This is a timing and important meta-analysis. The investigated topic is clinically important. The findings in this study is clinically relevant. I had only few comments to be addressed.

General comment:

There are so many grammatic errors (for example: The This study was registered with PROSPERO). Please find a native English speaker to resolve this issue.

Abstract:

  1. Please add one brief sentence to state the meta-analytic method used in this study in the section of abstract.
  2. Please spell out the “ WMD” and “95%CI” when they first appear.
  3. The authors said “The meta-analysis results showed that after tDCS treatment, the overall cognitive function of patients with AD and MCI was significantly improved (WMD=0.99; 95%CI, 0.32 to 1.66; P = 0.004),” in the abstract. I had to confirm what is the comparator for their effect size compared to? Placebo? Pre-post comparison in tDCS arm?

Introduction:

  1. The authors mentioned the different excitatory/inhibitory effect by the different polarity and also the different excitatory/inhibitory effect by the different current of tDCS. This is very excellent introduction. However, I would recommend the authors also to address another new concept in tDCS research, the neural noise theory. Please check the reference PMID: 21685932, and PMID: 33691622 and make a brief discussion about this.
  2. The authors had mentioned that there had been several meta-analyses published at present time. Some of those meta-analyses had provided promising result. The authors should provide a reason why the authors should do this new meta-analysis. Are the previous meta-analyses too poor or too small? Are there obvious heterogeneity in the previous meta-analyses? In addition, the authors should say clearly what new information will this new meta-analysis intend to provide.

Method:

  1. The authors followed the PRISMA guideline. This is great. However, since the latest PRISMA 2020 guideline had been published, I would recommend the authors to update to latest version. In addition, if the authors intend to update to 2020 version, the supplement table of “excluded studies and reason” would become a mandatory item in the latest version. Please provide this accordingly.
  2. I noticed that the authors had done subgroup analysis in the section of result. The authors should add a section of statement about their subgroup analysis process. Also, I would strongly recommend the authors also briefly mention the rationale of their cut-off point (i.e. number of stimulations ranged from 5-10-15, current density 0.06- 0.08-2.5).
  3. I recognized the authors did subgroup analysis based on current intensity and target region of stimulation. This is good. However, did the authors also consider to do subgroup analysis based on “excitatory or inhibitory” effect of tDCS?

Result:

  1. Here is a SERIOUS and MAJOR concern in the section of result. The authors said “The meta-analysis showed that compared with the control group, tDCS treatment significantly improved the overall cognitive function assessed by the MMSE of the experimental group, with a combined WMD of 0.99 (95%CI, 0.32 to 1.66; P=0.004; Fig.2a).” However, in the forest plot Fig 2a, the figure indicate “Favours Sham tDCS”. I am not sure what happen to this.
  2. Followed the comment 1, this mistake would lead me to not sure which direction of Fig 2b-2f to be correct. To right indicate Favours Sham tDCS or true tDCS?
  3. The authors choose to report publication bias with funnel plot. In the general issue, the funnel plot was reserved to be used in < 10 studies. In situation of > 10 studies, the better method of publication bias evaluation would be Egger’s test. Please address this issue.

Limitation:

  1. The authors said “The cognitive training methods of each study are different and may have a certain impact on the results”. I generally agreed with this statement. However, I would recommend the authors to do another test. The authors could arrange meta-analysis of placebo effect in the included RCTs (i.e. to test pre-post cognitive change in placebo group) so that the authors could provide further evidence to support or refute the statement of “The cognitive training methods of each study are different and may have a certain impact on the results”.
  2. The authors should make further statement about the heterogeneity among the included RCTs. Although not always achieve statistical significance, some heterogeneity could be detect through DIRECT observation. For example, the treatment duration, follow-up duration, ethnicity, blindness of study design, and so on.

Supplement materials:

  1. I am sorry but I did not see any supplement materials in the journal website.

Author Response

(The authors gave the same response as above.)

Round 2

Reviewer 1 Report

The paper makes much more sense with the included revisions. I am happy for it to be published in it's current form.

Reviewer 3 Report

The authors had addressed all my comments. The current version is good to be accepted.